# *OmniCache*: Multidimensional Hierarchical Feature Caching for Diffusion Models

## Abstract

Recent high-resolution image and video diffusion models (e.g., SD3, FLUX, Sora) have advanced generative intelligence but remain computationally expensive due to quadratic attention and multi-step inference. In this paper, we address the challenge of computational inefficiency in image & video generation by systematically exploiting inherent redundancy in intermediate representations. We identify four primary types of redundancy: intra-frame, inter-frame, motion, and step redundancy. To mitigate these, we propose *OmniCache*, a unified hierarchical caching framework that employs multidimensional feature reuse: Frame Cache, Block Cache, and Token Cache across transformer layers. These strategies enable us to compress spatial features in the temporal layers and temporal features in the spatial layers, significantly enhancing generation efficiency without the need for additional training. We further incorporate an orthogonal layered caching strategy to capture cross-step redundancy. We evaluate *OmniCache* on state-of-the-art diffusion models for both image and video generation, including SD3, SVD-XT, and Latte. It achieves up to 35% reduction in inference latency on Stable Diffusion 3 (SD3), 25% on SVD-XT, and 28% on Latte, while maintaining high visual fidelity. [1]

## 1 Introduction

With the rising popularity of Sora, VEO (Brooks et al., 2024; DeepMind, 2024), generating minute-long high-resolution videos has enabled users to bring their imagination to life. Producing long and coherent videos is a crucial step toward Artificial General Intelligence (AGI). Sora can generate 1800 frames of 1080p video in one run. However, as the resolution and frame rate increase, the required memory and computation rise exponentially, greatly elevating deployment costs.

Unlike image generation, video generation must ensure spatial continuity and vivid motion. To achieve this, methods such as Stable Video Diffusion (SVD) (Blattmann et al., 2023), VideoCrafter (Chen et al., 2024a), MicroCinema (Wang et al., 2023) and AnimateDiff (Guo et al., 2024) extend image diffusion models with temporal layers to ensure temporal consistency while generating each frame. Models like Latte (Ma et al., 2024a) and Open-Sora (Tech, 2023) utilize transformer-based Diffusion alternating between temporal attention and spatial attention at each layer. Consequently, the computational burden of generating N frames is linearly proportional to N, and doubling the video resolution quadruples the computational costs. However, this increase in frames and resolution leads to significant informational redundancy. For instance, as shown in Table 1, increasing frame rate from 1 FPS to 30 FPS raises bitrate threefold, while the frame rate increases thirtyfold; similarly, raising resolution from 360p to 1080p enlarges pixel area ninefold but bitrate only $3.55\times$. Thus, there exists substantial redundancy in the video generation process, and exploiting this redundancy to accelerate video production can significantly enhance efficiency.

We categorize the inefficiencies in diffusion-based video generation into four complementary types of redundancy:

---

[1]The code implementation will soon be made publicly available.

| Resolution & Frame Rate | Bit Rate (Kbps) |
|---|---|
| 640×360, 1 FPS | 192 |
| 640×360, 30 FPS | 576 |
| 1920×1080, 30 FPS | 2048 |

Table 1: Approximate H.265 recommended bit rates (Kbps) for various resolutions and frame rates (Hikvision, 2024).

- *Intra-frame redundancy*: Analogous to a codec's Intra Frame Compression, each frame inherently contains numerous repetitive patterns. By consolidating computations of similar regions within a frame, we can reduce the number of tokens required for spatial attention, thereby increasing computational efficiency.

- *Inter-frame redundancy*: Similar to a codec's Inter Frame Compression, the areas that change between frames are actually quite limited, with a substantial amount of redundant information. Consequently, codecs encode primarily the key frames, while other frames are derived using motion vectors to transmit information. Taking advantage of inter-frame redundancy, we can significantly enhance the generation efficiency via merging similar content across frames.

- *Motion redundancy*: In codecs, motion vectors are calculated on a block basis, primarily because the motion itself is inherently sparse. By merging temporal attention computations across different spatial positions, we can make the generation of motion more efficient.

- *Step redundancy*: In diffusion models, sampling cost increases with the number of denoising steps. We propose to exploit feature similarity across adjacent steps to cache computations, thereby improving denoising efficiency.

Prior studies have partially leveraged such observations. DeepCache, and Block cache Ma et al. (2023); Wimbauer et al. (2024) reuse feature maps across denoising steps in U-Net architectures, while Δ-Dit, ToCa, and Learning-to-Cache Chen et al. (2024b); Zou et al. (2025); Ma et al. (2024b) extend caching to transformer-based diffusion models. Other works explored low-resolution optical-flow-like representations (He et al., 2024) or image-to-video generation from single images (Yu et al., 2024; Ni et al., 2023), demonstrating the benefit of exploiting spatial and motion consistency. While effective, these approaches address only one redundancy dimension and lack a unified mechanism to coordinate and balance computation across spatial, temporal, and diffusion-step domains. In addition, ToMeSD (Bolya & Hoffman, 2023) accelerates Stable Diffusion by merging similar tokens via feature averaging, a design approach that is effective for image models but introduces challenges when extending to video diffusion, where preserving positional consistency across frames is critical

To address these limitations, we propose *OmniCache*, a unified hierarchical caching framework that removes redundancy across multiple levels of diffusion models. As shown in Figure 1, it integrates three feature-reuse modules—Token Cache, Frame Cache, and Block Cache, targeting intra-frame, inter-frame, and motion redundancy, respectively. It performs merging and unmerging through caching rather than averaging, preserving token order and positional consistency. [2] In addition, a complementary Layered Cache captures cross-step redundancy at the model-layer level. Together, these modules form a hierarchical coordinated system that enables compression and efficient computation reuse without modifying the model architecture or requiring retraining.

We formulate caching as an adaptive resource-allocation problem: Given the redundancy pattern across spatial, temporal, and diffusion-step dimensions, determine where caching yields the largest efficiency gain under a fixed-budget while maintaining perceptual quality. By combining lightweight similarity computation, patch-level hierarchical caching, and GPU-optimized Triton kernels, *OmniCache* achieves substantial inference acceleration with minimal perceptual quality loss.

The core contributions of *OmniCache* are:

---

[2]Throughout this paper, "merging" and "unmerging" denote caching-based feature reuse.

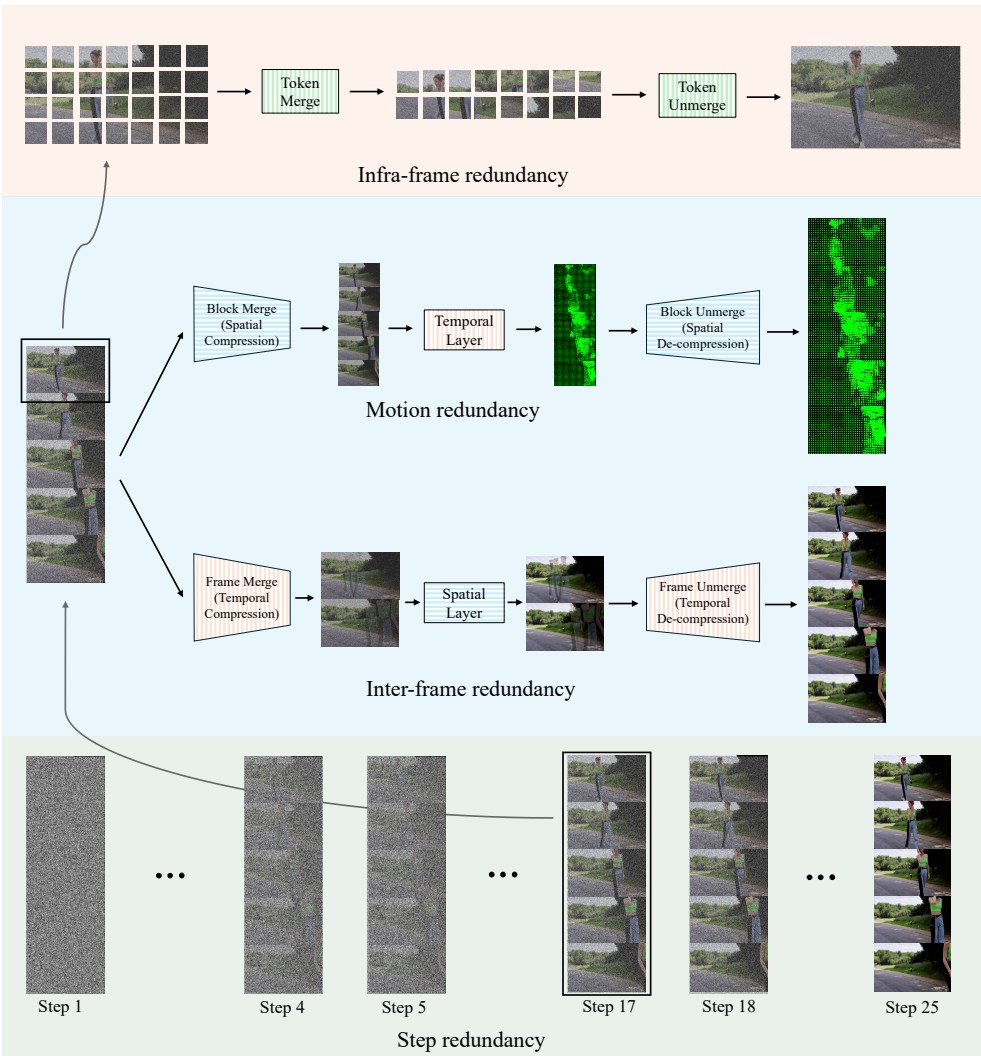

Figure 1: *OmniCache* reduces redundancy in a hierarchical cascade across time and space. First, it reduces step redundancy by caching features across consecutive denoising steps. For the remaining non-cached features, it exploits motion redundancy (repeated spatio-temporal patterns) and inter-frame redundancy (similarity across adjacent frames). Finally, within each frame, it identifies intra-frame redundancy by detecting repetitive tokens.

1. *Redundancy-first formulation for diffusion inference:* We introduce a unified perspective that characterizes inefficiencies in diffusion-based image and video generation along four complementary redundancy axes: intra-frame, inter-frame, motion, and denoising-step redundancy. By framing caching as an *adaptive resource allocation problem* across these dimensions and model layers, we provide a principled foundation for reducing redundant computations. Unlike prior work that targets a single redundancy type in isolation, this analysis directly guides where and how caching is applied across the model, forming the conceptual foundation of *OmniCache*

2. *Unified multi-granularity hierarchical caching framework:* We propose a unified caching framework that integrates: Frame Cache, Block Cache, Token Cache, and Layered Cache, each aligned with a specific redundancy type and architectural role. Through inference-time controls ($r_f$, $r_b$, $r_t$, $T$, $M$), our approach flexibly allocates compression across space, time, tokens, model layers to cache (caching scope), and caching interval across denoising steps, enabling fine-grained efficiency-quality tradeoffs.

3. *Structure-aware caching aligned with spatial-temporal layer roles:* A central design insight of *OmniCache* is that effective reuse depends on matching redundancy types to the architectural roles of layers. Specifically,

we cache spatial information within temporal layers to exploit appearance structure while preserving motion sparsity, and cache temporal information within spatial layers because these model appearance and are more tolerant to moderate temporal sub-sampling. Our S2T/T2S vs. S2S/T2T experiments demonstrate that mismatched compression leads to significant quality degradation, highlighting a structure-aware design principle absent from prior token-merging or step-caching methods.

4. *System-level design for real GPU inference acceleration.* We introduce patch-wise hierarchical caching to reduce similarity-computation overhead, layer and timestep-aware scheduling to better align reuse decisions with GPU execution, and custom Triton kernels that efficiently fuse merge and unmerge operations. These system-level optimizations ensure that caching translates into consistent wall-clock speedups on modern GPU backends. We validate this design on video diffusion models like SVD-XT (UNet-based) and Latte (Transformer-based), and image diffusion models like SD3, demonstrating consistent $25\% - 35\%$ inference speedups with minimal impact on visual quality and motion fidelity.

## 2 Related Works

### 2.1 Diffusion Models

Diffusion models (Ho et al., 2020; Dhariwal & Nichol, 2021) have demonstrated strong generative performance across both image and video synthesis, surpassing earlier GAN-based approaches. Early diffusion architectures predominantly relied on U-Net backbones for denoising; however, convolutional networks tightly couple positional information with feature maps, limiting scalability. Diffusion Transformers (DiT) (Peebles & Xie, 2023), which replace U-Nets with transformer architectures, underpin recent state-of-the-art text-to-image models such as Stable Diffusion 3 (Esser et al., 2024) and Flux (Black Forest Labs, 2025). These models typically employ low-resolution pretraining followed by resolution-specific finetuning, but inherit the quadratic cost of self-attention, making high-resolution generation increasingly expensive.

Stable Video Diffusion (SVD) (Blattmann et al., 2023) extends Stable Diffusion with temporal layers for image-to-video generation, while SV3D (Voleti et al., 2024) further adapts it for multi-view synthesis. Latte (Ma et al., 2024a) introduces a latent diffusion transformer for joint spatio-temporal modeling, while Sora (Brooks et al., 2024) and Open-Sora (Tech, 2023) further demonstrate the potential of transformer-based video diffusion. Collectively, these developments highlight the growing computational cost of modern diffusion models.

### 2.2 Efficient Diffusion Inference

Prior work reduces diffusion inference cost by exploiting redundancy in intermediate representations across denoising steps, layers, and tokens. DeepCache (Ma et al., 2023) observes that high-level semantic features in U-Net–based diffusion models change slowly across denoising steps and can be reused to reduce computation. Subsequent methods extend this idea to transformer-based diffusion models: $\Delta$-DiT (Chen et al., 2024b) accelerates DiT models by caching feature offsets and revealing block-specific roles, while ToCa (Zou et al., 2025) adaptively selects tokens to reuse based on temporal redundancy, error sensitivity, and layer depth. ToMe (Bolya et al., 2023) introduces bipartite token matching to merge similar tokens in ViT, while ToMeSD (Bolya & Hoffman, 2023) adapts the idea to diffusion models. While effective, these methods typically optimize reuse along a single axis (e.g., tokens, layers, or timesteps) and do not jointly coordinate feature reuse across interacting dimensions.

In contrast, *OmniCache* introduces a unified, hierarchical caching framework designed for both modern image and video diffusion models. We model redundancy across tokens, blocks, and timesteps and jointly adapt caching decisions across layers and denoising steps. This unified design naturally scales from image to video diffusion models.

### 2.3 Efficient Video Generation

CMD (Yu et al., 2024) proposes a content-motion latent diffusion model that encodes a video as a combination of a content frame (like an image) and a low-dimensional motion latent representation. This approach

enables efficient video generation, as it only requires generating a single content frame and a motion latent to reconstruct a vivid video. LFDM (Ni et al., 2023) use a diffusion network to predict optical flow to warp a user-provided image for video generation.

ToCa (Zou et al., 2025) and Learning-to-Cache (Ma et al., 2024b) focus on local reuse decisions within diffusion transformers: ToCa selects tokens to cache based on similarity and noise sensitivity, while Learning-to-Cache learns a timestep-dependent routing policy to cache entire transformer layers. Several video-specific inference accelerators further exploit spatio-temporal redundancy, including VidToMe (Li et al., 2024), which applies token merging for video editing scenarios, and BlockDance (Zhang et al., 2025), which reuses structurally similar spatio-temporal blocks in diffusion transformers. While effective, these approaches typically operate along a single redundancy axis and are tailored to specific architectural settings.

In contrast, *OmniCache* formulates caching as a global, hierarchical resource-allocation problem that coordinates reuse across multiple axes, including frame, block, token, and denoising-step-level redundancy. By integrating these mechanisms into a unified, budget-constrained inference-time framework, *OmniCache* enables flexible efficiency–quality tradeoffs across video diffusion models without requiring training or learned routing policies.

# 3 Methods

## 3.1 Background

**Diffusion Models.** Diffusion models generate data by iteratively denoising Gaussian noise through learned transformations. Given a timestep $t \in [1, T]$ and original data (or VAE latent) $x_0 \in \mathbb{R}^{C \times H \times W}$ for images or $x_0 \in \mathbb{R}^{N \times C \times H \times W}$ for videos, the forward process produces a noisy sample $x_t = \sqrt{\bar{\alpha}_t} x_0 + \sqrt{1 - \bar{\alpha}_t} \epsilon$, where $\epsilon \sim \mathcal{N}(0, 1)$ and $\bar{\alpha}_t$ follows a predefined noise schedule.

A noise prediction network $\epsilon_\theta(x_t, c, t)$ takes the noisy latent $x_t$, conditioning signal $c$ (e.g., text or reference image), and timestep $t$, and is trained to predict $\epsilon$ by minimizing the standard denoising objective (Ho et al., 2020), $\mathbb{E}\big[\|\epsilon - \epsilon_\theta(x_t, c, t)\|_2^2\big]$.

During inference, sampling starts from Gaussian noise $x_T$ and iteratively applies a solver to obtain $x_0$. For DDPM-style sampling (Ho et al., 2020), the update can be written as $x_{t-1} = \frac{1}{\sqrt{\alpha_t}}\Big(x_t - \frac{\beta_t}{\sqrt{1-\bar{\alpha}_t}}\epsilon_\theta(x_t, c, t)\Big) + \sigma_t z$, with $z \sim \mathcal{N}(0, I)$. While modern solvers (Song et al., 2020; Lu et al., 2022; 2025; Karras et al., 2022) vary in formulation, they all depend on repeated evaluations of $\epsilon_\theta$ across timesteps.

Unlike image diffusion, video diffusion must additionally preserve frame-to-frame coherence. To balance spatial detail and temporal consistency, modern video diffusion models alternate spatial and temporal attention within U-Net or transformer blocks, capturing motion without incurring prohibitively large attention costs.

**Token Merge and Unmerge.** Token Merging (ToMe) (Bolya et al., 2023) improves Vision Transformer efficiency by merging redundant tokens via bipartite similarity matching. For diffusion models, where every spatial position must estimate noise, ToMeSD (Bolya & Hoffman, 2023) adapts this idea by inserting merge and unmerge operations around attention and feed-forward layers to reduce intermediate computation while preserving feature-map resolution.

ToMeSD partitions tokens into source and destination sets, matches similar pairs based on cosine similarity, and merges selected tokens via feature averaging. During unmerging, the averaged features are broadcast back to the original token positions. While effective for image diffusion, this averaging-based merge reorders sequence elements and disrupts order-sensitive positional encodings (e.g., RoPE), and the unmerge operation lacks explicit positional consistency, limiting its applicability to modern video diffusion models.

To address these limitations, we introduce *Token Cache*, which also uses bipartite matching to identify redundant tokens but avoids averaging. Instead, Token Cache deterministically retains one token and discards its matched counterpart, preserving sequence order. During unmerging, positionally consistent features are reused from previous denoising steps, leveraging step redundancy to maintain spatial coherence while reducing redundant computation.

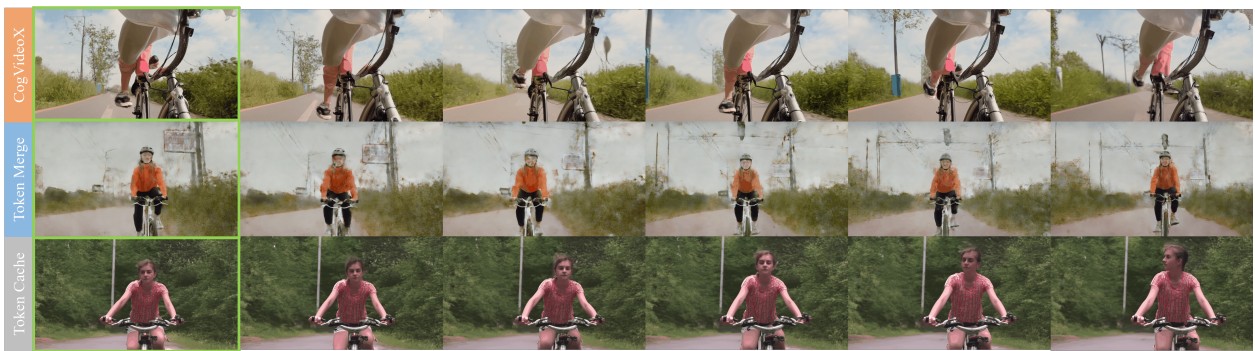

Figure 2: Limitation of token merging.

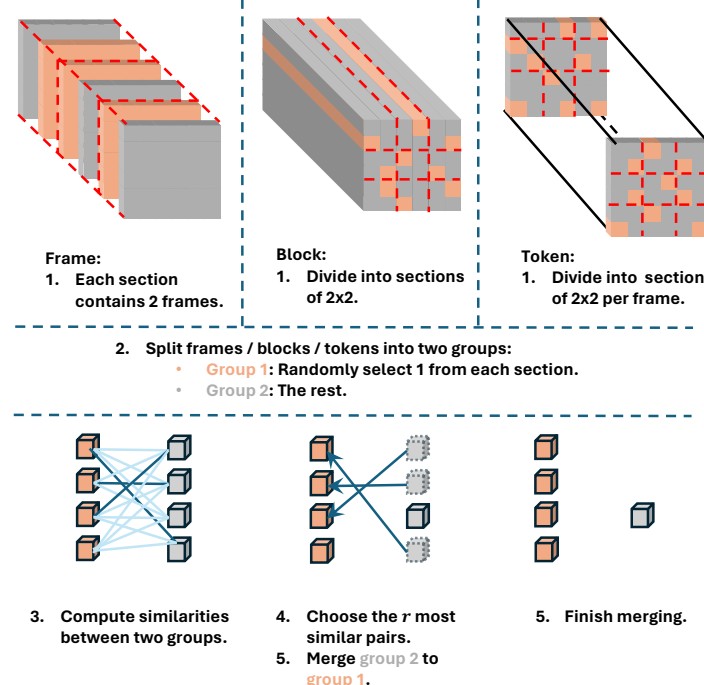

Figure 3: Illustration of *OmniCache*'s *Frame Cache*, *Block Cache*, and *Token Cache* operations. We designed three dimensions of merge operations for modern video diffusion models to more flexibly reduce feature redundancy and enhance efficiency.

## 3.2 *OmniCache*

We present *OmniCache*, a unified hierarchical caching framework that removes redundancy across multiple levels of diffusion models. As shown in Figure 1, it integrates Token Cache, Frame Cache, and Block Cache to address intra-frame, inter-frame, and motion redundancy, respectively, together with a complementary Layered Cache for cross-step reuse. All cache modules follow a shared design principle: reuse intermediate features where redundancy is high and approximation tolerance is large under an explicit inference-time budget. Rather than applying caching decisions independently, *OmniCache* formulates feature reuse as a hierarchical resource-allocation problem across multiple axes, including space, time, tokens, layers, and denoising steps, enabling coordinated computation reuse that improves efficiency while preserving perceptual quality, without modifying the model architecture or requiring retraining.

### 3.2.1 Motion Redundancy and Block Cache

In video diffusion models, motion features are typically extracted by dedicated temporal transformer layers. For an input feature map of dimensions (B, T, C, H, W), we reshape it to (B*H*W, T, C) to compute temporal features. This reshaping allows us to calculate temporal relationships across T frames at each H and W coordinate of the video. We refer to this sequence of 1*1*T attention input as a block. We believe that a significant portion of motion in a video is repetitive, such as background shifts due to camera movement or rigid body motion. This redundancy is why codecs like H.265 use block-wise motion vectors to encode movement across entire regions. Therefore, within the temporal transformer's input blocks, there is substantial potential for computation aggregation.

We cannot directly use ToMeSD's method here, as it only calculates similarity and merges tokens within a single image. However, inspired by its efficient bipartite graph matching to find the closest edges, we can apply a similar approach to compute similarity between blocks for Block Cache.

To calculate the similarity between blocks, the simplest method is to concatenate the features of all T elements in a block. As shown in Figure 3, we transform the input features to (B, H*W, T*C), isolating these H*W blocks, each containing T*C features. Following ToMeSD, we randomly select one element from a 2*2 block as the source block, with the remaining elements serving as destination blocks. We then compute the similarity between the source and destination blocks by multiplying the source vector by the transpose of the destination vector. Instead of averaging similar blocks, we skip the $r_b$ most similar **src** blocks during the merging process and reuse the cached features from the previous denoising step to reconstruct the features in the current step during unmerging. This strategy effectively reduces motion redundancy while preserving positional consistency.

### 3.2.2 Inter-frame redundancy and Frame Cache

As we know, adjacent frames in a video often contain very close content, such as a static background or minor motion differences. In works like Ni et al. (2023), the warp method is used to efficiently generate videos based on overlapping images. This inspires us to consider whether merging some similar feature maps in the spatial layer during the generation process, allowing them to share computation, could take advantage of this characteristic of videos.

Again, we utilize the bipartite graph matching algorithm to find edges between closest spatial content between frames for Frame Cache. For an input feature map of dimensions (B, T, H, W, C), we reshape it to (B, T, H*W*C) to concatenate all features of each frame as the feature of that frame. In this case, we divide these $T$ frames into **src** and destination **dst** sets. To balance randomness and uniformity, we select one frame between every two adjacent frames to enter the **dst** frame set, and then select the $r_f$ most similar frames from the **src** frames. Similar to Block Cache, the most similar $r_f$ frames from **src** set are skipped, and the cached features from the previous denoising step are reused.

A direct concern is whether this approach reduces the frame rate of video generation. In fact, our computation in the motion layers still uses the original frame rate input, only merging some similar feature map calculations in the spatial layers. The independent motion relationships generated in the motion layers at the original frame rate can still ensure the richness of frame-to-frame transitions in the generated video. Therefore, the similar frames averaged in the spatial layer will regain diversity after being processed in the motion layer.

### 3.2.3 Intra-frame redundancy and Token Cache

Despite performing Frame Cache, there remains considerable redundancy among tokens within each frame. For instance, in two similar frames depicting the sky and the beach, there will still be substantial spatial redundancy after Frame Cache. Therefore, we apply Token Cache to each frame before entering the spatial layers.

The input to Token Cache is the feature map generated from Frame Cache, but this leads to a reduced frame rate for video generation where the generated video has fewer motions. To address this, we first calculate the similarity between all frames without merging the similar ones, then split them into merged and unmerged

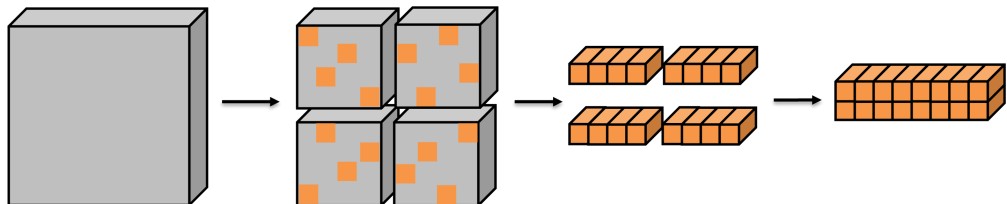

Figure 4: Illustration of Hierarchical Caching

feature maps. For merged feature maps, we identify $r_f$ pairs of frames and merge similar input tokens in each pair to reduce intra-frame redundancy. The processed feature map will have the shape (B, $r_f$*2, H, W, C) which we reshape to (B*$r_f$, 2*H*W, C). Within 2*H*W tokens, we select two mutually exclusive elements from each 2*2*2 cube and merge them into H*W-$r_t$ tokens. For unmerged feature maps, within the H*W region of each frame, we randomly select one element from each 2*2 block as **src** tokens, while the remaining elements serve as **dst** tokens. We then merge the $r_t$ most similar tokens. Finally, the results from merged and unmerged feature maps are concatenated. This approach maintains the original frame rate while merging similar tokens, minimizing temporal information loss compared to the sequential operation of Frame Cache.

### 3.3 Hierarchical Caching

Hierarchical caching provides a practical mechanism to implement feature reuse at multiple spatial scales by controlling the scope at which similarity computation and caching are applied. For Frame Cache, the input feature map has the shape (B, T, H*W*C), which results in an excessively large dimension when computing similarity due to the large size of the spatial content. To reduce overhead, we divide the spatial content into multiple patches, resulting in a shape of (B, T, k, H/k, k, W/k, C) where k*k is the number of patches, shown in Figure 4. We then permute and reshape the feature map to (B*k*k, T, (H/k)*(W/k)*C) for similarity computation. Using patches for Frame Cache reduces overhead by over 20× for Stable Video Diffusion (SVD) without degrading quality or motion. Similarly, patches can be applied to Block Cache and Token Cache, as adjacent pixels are more likely to be merged. This process reduces the overhead of Block Cache and Token Cache by about 18× and 15×, respectively, in SVD. The usage of patches also enhances caching performance by enabling patch-to-patch caching, which allows for more flexible and diverse caching options across different patches between frames.

After performing caching with patches, we can increase the patch size to allow for further caching or revert the patches back to frames for global caching. This enables a hierarchical caching strategy, where we adjust the caching ratio and hierarchical level to optimize performance.

**Implementation Efficiency**  To ensure that hierarchical caching translates into real wall-clock speedups, we implement key merge and un-merge operations using custom Triton kernels. These kernels optimize gather–scatter patterns and significantly reduce runtime overhead; detailed kernel design and profiling results are provided in Appendix A.

### 3.4 Step Redundancy and Layered Cache

Prior work such as DeepCache and Δ-DiT (Ma et al., 2023; Chen et al., 2024b) has shown that feature distributions within individual layers evolve slowly across adjacent denoising steps, making cross-step feature reuse effective for accelerating diffusion inference. These methods identify cacheable layers or blocks based on observed feature similarity, and apply reuse uniformly across selected steps.

In contrast, we formulate step-level caching as a *data-centric, budget-constrained scheduling problem* that explicitly balances efficiency gains against quality degradation across both layers and denoising timesteps. This formulation allows Layered Cache to complement our Frame, Block, and Token Cache within a unified hierarchical framework.

Specifically, for a given diffusion model, we profile each transformer or U-Net layer over a range of denoising steps using prompts of varying length from the MS-COCO 2017 dataset. For each layer–timestep pair $(l, t)$, we measure (i) the average execution latency and (ii) the resulting quality degradation when caching is applied at that location, quantified using perceptual metrics such as FID, Inception Score, or FVD. We then assign each $(l, t)$ pair a weight $w_{(l,t)} = \frac{\Delta \text{quality}(l,t)}{\text{latency}(l,t)}$, which represents the quality loss incurred per unit of saved computation.

Given a user-defined target speedup $s_r$ and cache interval $T$, we construct an initial cache schedule by greedily selecting layer-timestep pairs with the lowest $w_{(l,t)}$, i.e., those that offer the most favorable tradeoff between efficiency and quality. This procedure can be interpreted as a knapsack-style approximation that prioritizes cache placements with high benefit-to-cost ratios. The resulting cache proposal is summarized in Algorithm 1.

To ensure correctness and avoid redundant computation, we further refine the initial cache schedule using the model's computation graph. By explicitly tracking dependencies between cached outputs across layers and timesteps, we remove unnecessary cache operations while preserving execution order and data availability. This dependency-aware refinement is described in Algorithm 2, with full details provided in Appendix B.

---

**Algorithm 1** Cache Schedule Proposal

**Require:** weights $w_{(l,t)}$, latencies, target speedup $s_r$, caching interval $N$
1  **function** GET_CACHE_STRATEGY
2      sort($w$) by $w_{(l,t)}$
3      $TimeSave \leftarrow 1 - \frac{1}{s_r}$; $sum \leftarrow 0$; $i \leftarrow 0$
4      **while** $sum < TimeSave$ **do**
5          $sum \leftarrow sum + w[i][l]$
6          $CacheStrategy[w[i][layer], w[i][timestep]] \leftarrow 1$
7          $i \leftarrow i + 1$
8      **end while**
9      **return** $CacheStrategy$
10 **end function**

---

**Algorithm 2** Cache Strategy Refinement

**Require:** Model graph $\mathcal{G}$, CacheStrategy $\mathcal{C}$
1  **Preprocess:** Initialize metadata for nodes
2  **Build Dependencies:** Set child/parent relations
3  **Count Dependencies:** Compute child_num, parent_num
4  **Prune Redundancies:** Remove unnecessary cache steps
5  **Prune Consecutive Steps:** Drop identical outputs
6  **Recalculate Counters:** Update dependency counts
7  **Align Steps:** Match with parent cache intervals
8  **return** Optimized strategy $\hat{\mathcal{C}}$

---

Figure 5: **(Left)** Algorithm for generating cache schedule proposals. **(Right)** Refinement process for optimizing cache strategies.

## 4 Experiments

### 4.1 Experimental Settings

**Model Configurations.** We evaluate *OmniCache* on representative image and video diffusion models, including Stable-Video-Diffusion (SVD-XT) for image-to-video generation and Stable-Diffusion-3-medium (SD3-medium) Esser et al. (2024) for text-to-image generation using NVIDIA A100 with 40 GB VRAM. Similar to ToMe, we utilize these models directly with our method without training. SD3-medium employs a Flow Matching Euler Discrete scheduler with 28 sampling steps, while SVD-XT uses an Euler Discrete sampler with 25 steps. Unless otherwise stated, SVD generates 25 frames at $576 \times 1024$ resolution, and Latte-1 T2V generates 16 frames at $512 \times 512$ resolution.

**Evaluation and Metrics.** For image-to-video evaluation, we generate 10,000 videos with SVD on UCF101 dataset and report FVD scores. For text-to-image evaluation, we generate images from 8,000 randomly sampled MS-COCO'14 captions and compute FID score using 20,000 real images, while CLIP score is used to measure image-to-text alignment.

### 4.2 Design Choices

We investigate different caching ratios and layer selections to identify where hierarchical caching improves efficiency without degrading quality. Across all models, early and late layers are consistently more sensitive to feature reuse, aligning with prior observations that they encode coarse structural and fine-grained details, respectively.

For Stable Video Diffusion (SVD), we find that the final spatial transformer layers are particularly sensitive to frame caching. Accordingly, we exclude the last decoder layer from Frame Cache and avoid applying Frame or Token Cache to FFN layers, which consistently degrades visual quality. Under these constraints, we observe a stable operating regime with caching ratios up to approximately $r_f \leq 50\%$, $r_b \leq 40\%$, and $r_t \leq 50\%$, beyond which over-smoothing and motion artifacts begin to appear. For Latte, we exclude caching in the first and last transformer layers and apply uniform caching ratios of $r_f = 30\%$, $r_b = 40\%$, and $r_t = 20\%$ across the remaining layers. We further observe that applying Frame or Token Cache to FFN layers leads to noticeable degradation, and therefore restrict caching to attention blocks only. For SD3-medium, we apply hierarchical token caching to both attention and FFN layers, but exclude the top and bottom four transformer blocks, which are highly sensitive to caching. This behavior aligns with observations in $\Delta$-DiT (Chen et al., 2024b), where early and late blocks play critical roles in coarse and fine feature synthesis. Across models, our ablations suggest that conservative caching ratios (approximately $r_f \leq 40\%$, $r_b \leq 30\%$, $r_t \leq 40\%$) provide a robust quality–efficiency tradeoff, while more aggressive settings increase the risk of artifacts.

Detailed quantitative results for different ratio settings are reported in Tables 2 and related ablation tables. Based on these findings, we recommend the above conservative regimes as default settings for practical deployment.

### 4.3 Main results

**Training-free improvements on state-of-the-art image and video Diffusion models.** We evaluate *OmniCache* on UNet–based image-to-video models (e.g., SVD) and DiT-based models for text-to-video (Latte) and text-to-image (SD3-medium) . As shown in Figures 6, 7, and 8, *OmniCache* achieves nearly lossless generation quality, while reducing inference latency by 25% on SVD, 28% on Latte and 35% on SD3. At matched compression rates, our method consistently produces higher-quality outputs than TomeSD-based baselines.

| Method | Latency and Speedup | | | Evaluation Metrics | |
|---|---|---|---|---|---|
| | $S_{DiT}$ ↑ | $L_{DiT}$ (ms) | Retrain | FID ↓ | CLIP ↑ |
| SD3-medium | 1.00 | 138.36 | ✗ | 31.57 | 18.60 |
| Omnimerge (cache) 40% | 1.20 | 114.96 | ✗ | 31.42 | 20.26 |
| Omnimerge (cache) 50% | 1.27 | 109.00 | ✗ | 31.94 | 20.32 |
| Omnimerge (cache) 60% | 1.35 | 102.88 | ✗ | 32.45 | 20.38 |
| Layer-cache | 1.75 | 79.00 | ✗ | 31.36 | 20.11 |
| Omnimerge (cache)+Layer-cache 40% | 2.28 | 60.55 | ✗ | 31.14 | 20.28 |
| Omnimerge (cache)+Layer-cache 50% | 2.32 | 59.71 | ✗ | 31.95 | 20.30 |

Table 2: **Quantitative comparison of text-to-image generation** on MS-COCO2014 with SD3-medium. DiT latency is reported in milliseconds.

**Quantitative Metrics for SVD.** For SVD, we adopt the best-performing configuration with $r_f = 50\%$, $r_b = 40\%$, and $r_t = 50\%$. As shown in Table 3, *OmniCache* achieves FVD scores nearly identical to the non-caching baseline while reducing inference latency by 25 %, all in a training-free manner.

For comparison, we evaluate TomeSD using its best-performing setting with 40% token merging. While TomeSD improves efficiency, *OmniCache* better preseves FVD and delivers an additional 11% speedup, demonstrating a stronger quality-efficiency tradeoff than single-dimension token merging approaches.„

**Spatially Compressing Temporal and Temporally Compressing Spatial Layers Minimizing Information Loss.** As shown in Figure 1, we hypothesize that compressing the spatial dimension in the temporal layer and the temporal dimension in the spatial layer results in less information loss. We conducted experiments to test this hypothesis by swapping the positions of frame caching (caching 30% of frames) and block caching (caching 30% of blocks). As depicted in Figure 9, the results of Spatially compressing Temporal layer (S2T) and Temporally compressing Spatial layer (T2S) closely resemble the original

| Method | Ratio (%) | FVD ↓ | s/im ↓ |
|---|---|---|---|
| Ground Truth | 0 | 502.68 | 110 |
| Token Merge (ToMeSD) | 40 | 503.8 | 92.6 |
| *OmniCache* | / | 503.12 | 82.4 |
| Frame Cache | 20 | 489.14 | 105 |
|  | 40 | 495.89 | 99.3 |
|  | 60 | 535.07 | 94.8 |
| Token Cache | 20 | 497.49 | 102 |
|  | 40 | 502.5 | 94.3 |
|  | 60 | 517.64 | 84 |
| Block Cache | 10 | 499.84 | 107 |
|  | 30 | 502.71 | 103 |
|  | 50 | 528.24 | 98.8 |

Table 3: Quantitative evaluation for different caching ratios for SVD.

SVD results. However, Spatially compressing the Spatial layer (S2S) and Temporally compressing the Temporal layer (T2T) resulted in a noticeable decrease in frame rate, as well as the appearance of blurriness and artifacts.

***OmniCache* Enables More Efficient Feature Compression.** We further demonstrate the effectiveness of *OmniCache* in feature compression under the S2T and T2S strategies, where Frame Cache, Block Cache, and Token Cache jointly minimize information loss. As shown in Figure 10, we compare our caching operations against direct interpolation baselines at a 40% compression ratio. Frame Interpolation (FI) leads to a significant drop in frame rate, almost to the point of stalling. Even when the temporal dimension is compressed in the Spatial layer, direct interpolation still results in considerable information loss. Block Interpolation (BI) yields very blurry results, with direct Spatial interpolation in the temporal layer leading to a degraded motion fidelity due to incorrect temporal information between frames Similarly, Token Interpolation (TI) causes noticeable spatial blurring when compressing intra-frame content. In contrast, all three caching methods in *OmniCache* yield results closely matching the original SVD. These results highlight the advantage of caching-based reuse over direct interpolation for exploiting video redundancy efficiently.

### 4.4 Ablations

**Analysis of Different Caching Compression Ratios.** As we analyze in Table 3, the effects of frame caching, block caching, and token caching are quite interesting. We observe that within 40% for frame caching, 40% for token caching, and 30% for block caching, there is no decrease in the FVD scores; in fact, there is even a slight improvement. This demonstrates the effectiveness of our method in removing video redundancy.

**Analysis of Speedup Ratios & Output Quality with Layered Caching.** To analyze the effectiveness of our Layered Caching algorithm, we set up a caching interval $N = 5$ and computed cache schedules for various speedups for Stable-Diffusion-3. For baseline, we used DeepCache's approach with caching intervals ($N = 2, 3$). We measured the model's end-to-end speedup for computational efficiency while perceptual (FID,CLIP) and Pixel-wise (LPIPS, SSIM, PSNR) for output quality by generating 2000 images using random image-text pairs from MS-COCO 2017 dataset.

From our analysis, we can observe that for same speedups, our approach outperforms baseline's performance. Moreover, with increasing speedup values, the model's output quality remains quite close to the original model's performance.

| Method | Speedup | Perceptual | | Pixel-wise | | |
|---|---|---|---|---|---|---|
| | | FID ↓ | CLIP ↑ | LPIPS ↓ | SSIM ↑ | PSNR ↑ |
| **SD3 (w/o caching)** | 1.000x | 39.187 | 32.169 | 0.000 | 1.000 | inf |
| **DeepCache** (int=2) | 1.8928x | 39.0282 | – | – | – | – |
| DeepCache (int=3) | 2.5552x | 39.6342 | – | – | – | – |
| **Our Method** | 1.645x | 37.546 | 32.096 | 0.329 | 0.634 | 14.801 |
| | 1.811x | 37.222 | 32.250 | 0.279 | 0.681 | 16.333 |
| | 1.889x | 36.598 | 32.027 | 0.367 | 0.597 | 13.955 |
| | 1.969x | 36.852 | 31.993 | 0.367 | 0.598 | 13.968 |
| | 2.184x | 35.353 | 31.676 | 0.363 | 0.596 | 14.680 |
| | 2.348x | 35.560 | 31.579 | 0.362 | 0.596 | 14.617 |
| | 2.506x | 35.619 | 31.676 | 0.346 | 0.615 | 15.053 |
| | 2.551x | 35.979 | 31.511 | 0.364 | 0.596 | 14.672 |
| | 2.655x | 35.991 | 31.207 | 0.374 | 0.584 | 14.626 |
| | 2.778x | 36.202 | 31.178 | 0.376 | 0.583 | 14.597 |

Table 4: Performance metrics of Stable-Diffusion-3 for different speedups. Image metrics are computed using 2000 random image–text pairs from the MS-COCO 2017 dataset.

## 5 Limitations

While our work focuses on training-free inference-time acceleration, the redundancy analysis underlying *OmniCache* may also be beneficial for training and fine-tuning diffusion models. For example, step-reduction techniques such as Progressive Distillation Salimans & Ho (2022) could be synergistically combined with our approach to jointly reduce the number of denoising steps and the per-step computation cost, potentially yielding multiplicative efficiency gains.

*OmniCache* introduces four granular caching mechanisms: Frame Cache, Block Cache, Token Cache, and Layered Cache, whose effectiveness depends on the choice of caching ratios and the amount of redundancy present in the input content. In scenarios with limited redundancy, such as videos with extremely rapid, non-repetitive motion or prompts requiring globally varying fine details at every frame, aggressive caching may lead to over-smoothing or degraded motion fidelity. Our ablation studies identify conservative operating regimes in which speedups increase while perceptual quality and motion coherence remain largely unchanged; however, the cache scheduling strategy remains heuristic and does not claim global optimality. While we formulate caching as a budget-constrained resource allocation problem guided by a principled benefit–cost metric; reinforcement learning or black-box optimization strategies could lead to more adaptive or content-aware scheduling strategies (e.g., learned or prompt-dependent policies) thereby further improving robustness and performance, which we leave for future work.

## 6 Conclusion

In summary, *OmniCache* presents a unified and effective approach to addressing the computational inefficiencies of video diffusion by explicitly modeling and exploiting four complementary sources of redundancy: intra-frame, inter-frame, motion, and denoising-step redundancy. By coordinating Frame Cache, Block Cache, Token Cache, and Layered Cache within a hierarchical caching framework, *OmniCache* integrates multiple reuse mechanisms into a single inference-time system, enabling substantial efficiency gains without modifying model architectures or requiring retraining.

Our experimental results demonstrate the robustness and generality of *OmniCache* across state-of-the-art open-source diffusion models, including SVD-XT, Latte, and SD3. In practice, *OmniCache* achieves training-free inference speedups of 25% on SVD-XT, 28% on Latte, and 35% on SD3, while preserving visual quality and motion fidelity, highlighting its practical value for scalable image and video generation.

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

## Appendix

## A   Triton Kernel Acceleration

To improve the efficiency of the merging and unmerging operations, we replace the original implementation with custom Triton kernels. These kernels leverage fine-grained GPU parallelism and memory coalescing to optimize the gather–scatter pattern inherent to our token permutation process. Specifically, we design block-level parallel kernels that operate on batched token indices, performing atomic accumulation and permutation across the spatial and temporal dimensions. By controlling tile sizes, warp scheduling, and staged memory access, the Triton kernels achieve high throughput while avoiding redundant global memory synchronization. This design enables direct in-GPU reduction (e.g., mean aggregation) without host round-trips, significantly reducing latency. In our profiling, the Triton-based implementation yields substantial speedups over the original version, demonstrating the benefit of custom kernel fusion and memory-efficient accumulation in our caching pipeline. Additional experiments evaluating the performance of our Triton kernels compared to PyTorch implementations on SD3-medium, across varying token group sizes in the hierarchical caching setup, are discussed below.

### A.1   Performance Evaluation of Triton Kernels

To further analyze the performance of our custom Triton kernels, we compare them against the baseline PyTorch implementation on the SD3-medium model under varying number of token groups $(1, 2, 4, 8)$ configurations in the hierarchical caching pipeline. Moreover, as described in Section 4.2, we omitted the inital and final 4 transformer blocks as they are sensitive to hierarchical caching which in turns impacts the generated image quality. As shown in Figure 11, the Triton-based kernels consistently outperform the PyTorch counterparts achieving $> 2.0\times$ speedup for merge and unmerge routines, demonstrating superior scalability and reduced latency as the number of token groups increases.

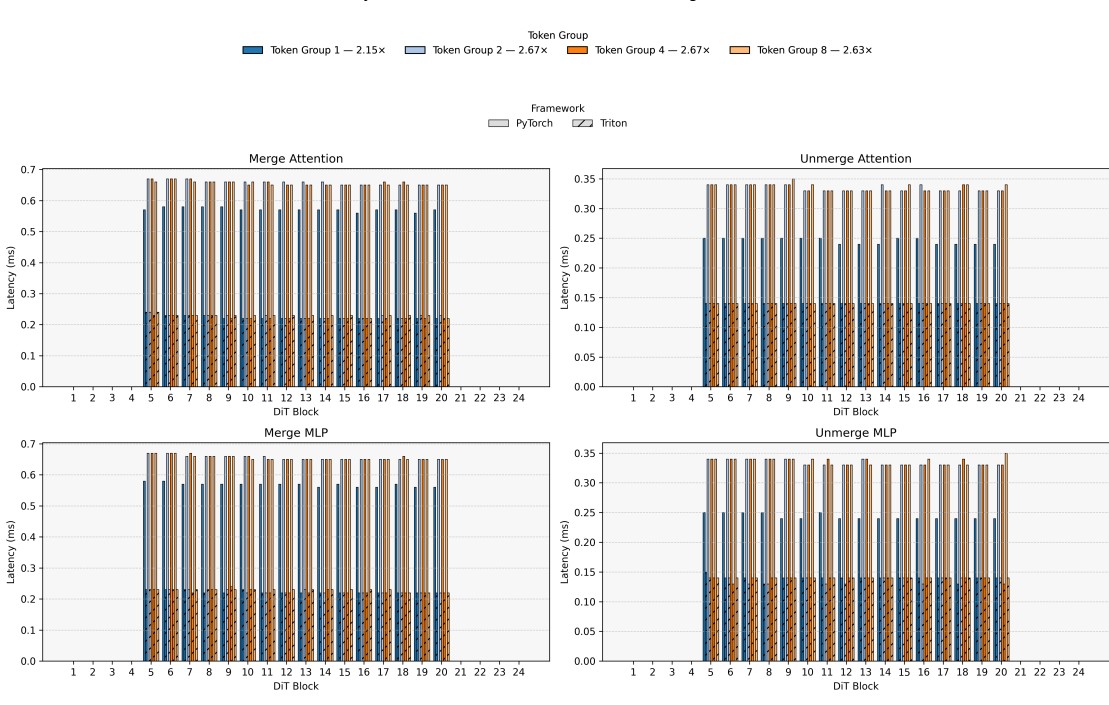

Figure 11: Comparison of our custom Triton kernel implementation against the PyTorch baseline on SD3-medium under different token group sizes in hierarchical caching. The Triton kernels achieve higher throughput and lower latency due to optimized memory access and fused operations. Average speedups for different token groups is provided alongside the Token Group legend

# B   Cache Strategy Refinement Algorithm

For completeness, we include the full pseudocode of the cache strategy refinement procedure introduced in Section 3.4. This algorithm refines the preliminary cache schedule proposed in Algorithm 1 by pruning redundant cache steps, realigning dependencies, and optimizing the caching intervals for computational efficiency.

---

**Algorithm 3** Cache Strategy Refinement Algorithm

---

**Require:** Model Compute Graph $\mathcal{G}$, CacheStrategy $\mathcal{C}$
  **Preprocess:**
  **for** each node in cachelist **do**
    Initialize metadata (children, parents, flags, counters) for the node
  **end for**
  **Build Dependency Graph:**
  **for** each node **do**
    Add child and parent relationships based on model structure
  **end for**
  **Calculate Dependency Counters:**
  **for** each node **do**
    Set child_num and parent_num based on relationships
  **end for**
  **Remove Redundant Cache Steps:**
  **for** each node in topological order **do**
    Use flags to identify and remove unnecessary cache steps
    Update flags for child nodes
  **end for**
  **Remove Consecutive Cache Steps with Identical Outputs:**
  **for** each eligible node in reverse topological order **do**
    **for** each pair of consecutive cache steps **do**
      **if** outputs are identical **then**
        Remove the redundant cache step
      **end if**
    **end for**
  **end for**
  **Recalculate Dependency Counters**
  **Align Cache Steps with Parent Dependencies:**
  **for** each node in topological order **do**
    **for** each cache step not in interval **do**
      Update to minimum parent cache step $\geq$ current step
    **end for**
  **end for**
  **Return** Optimized CacheStrategy $\hat{\mathcal{C}}$

---

This refinement process enforces structural consistency within the model's compute graph, eliminating redundant computations while maintaining temporal and dependency integrity across cached layers.

# C   Broader Societal Impact

Below we discuss the societal impact of *OmniCache* which lies in improving accessibility, sustainability, and deployment efficiency of existing generative systems.

**Energy Efficiency and Environmental Impact.**   By reducing inference latency by approximately 25–35% across multiple diffusion models, *OmniCache* proportionally reduces energy consumption for a fixed hardware configuration. For example, in our SVD-XT experiments, generating a 25-frame $576 \times 1024$ video requires approximately 110 seconds on a single $300\,\text{W}$ GPU in the baseline setting, corresponding to about $9.2\,\text{Wh}$ per generation. With *OmniCache*, latency is reduced to 82.4 seconds, lowering energy usage to approximately $6.9\,\text{Wh}$ per generation, a savings of about 25%. At a scale of 1M video generations per day, this corresponds to roughly $2.3\,\text{MWh}$ of energy saved per day, or approximately $840\,\text{MWh}$ annually. Assuming a conservative grid emission factor of $0.4\,\text{tCO}_2/\text{MWh}$, this translates to an estimated reduction of about $336\,\text{tCO}_2$ per year. While exact values depend on hardware and energy sources, this example illustrates the potential environmental benefit of inference-time efficiency improvements at scale.

**Accessibility and Cost Reduction.** Because *OmniCache* is training-free and operates entirely at inference time, it can be applied to pre-trained diffusion models on commodity GPUs without additional fine-tuning. This lowers the hardware and computational barriers for individual creators, small studios, and researchers, enabling broader access to high-quality image and video generation tools. For cloud-based services, inference speedups translate into reduced GPU-hour consumption, which may lower operational costs and, in turn, reduce costs passed on to end users.

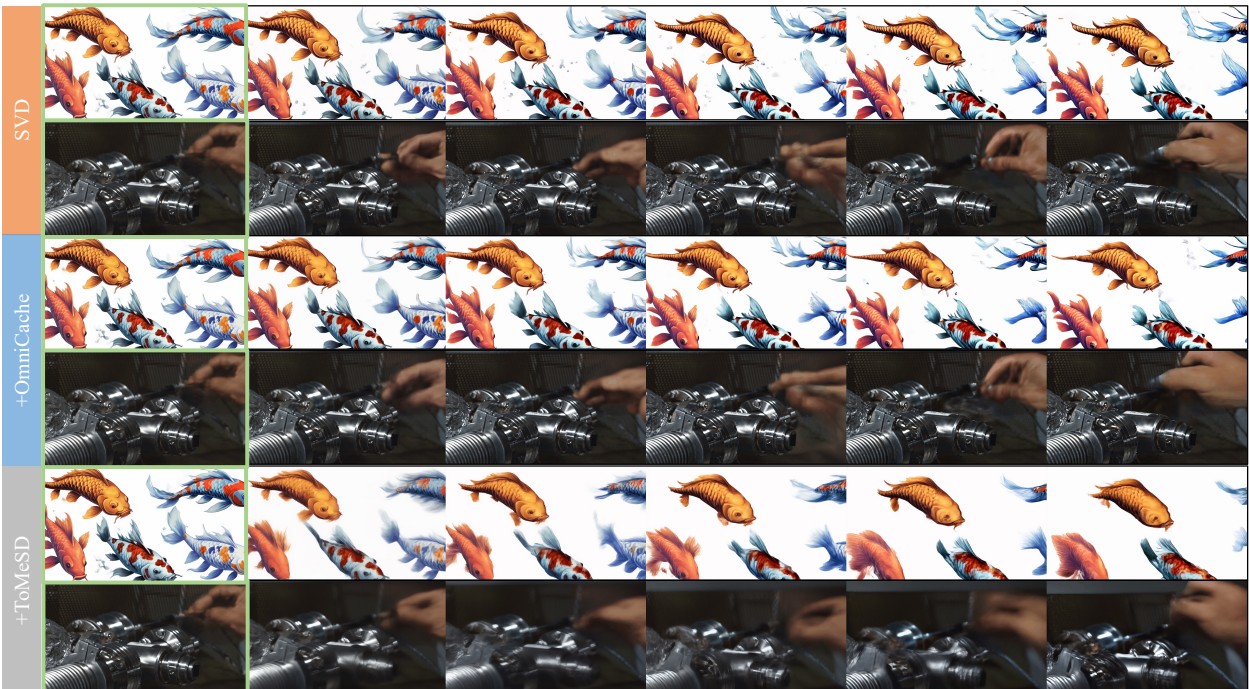

Figure 6: Performance of *OmniCache* on SVD. With $r_f$, $r_b$, $r_t$ set to (50%, 30%, 50%), compared against TomeSD with $r_t = 65\%$ at the same compression rate. We achieve similar performance to the original SVD while improving inference time by 25%.

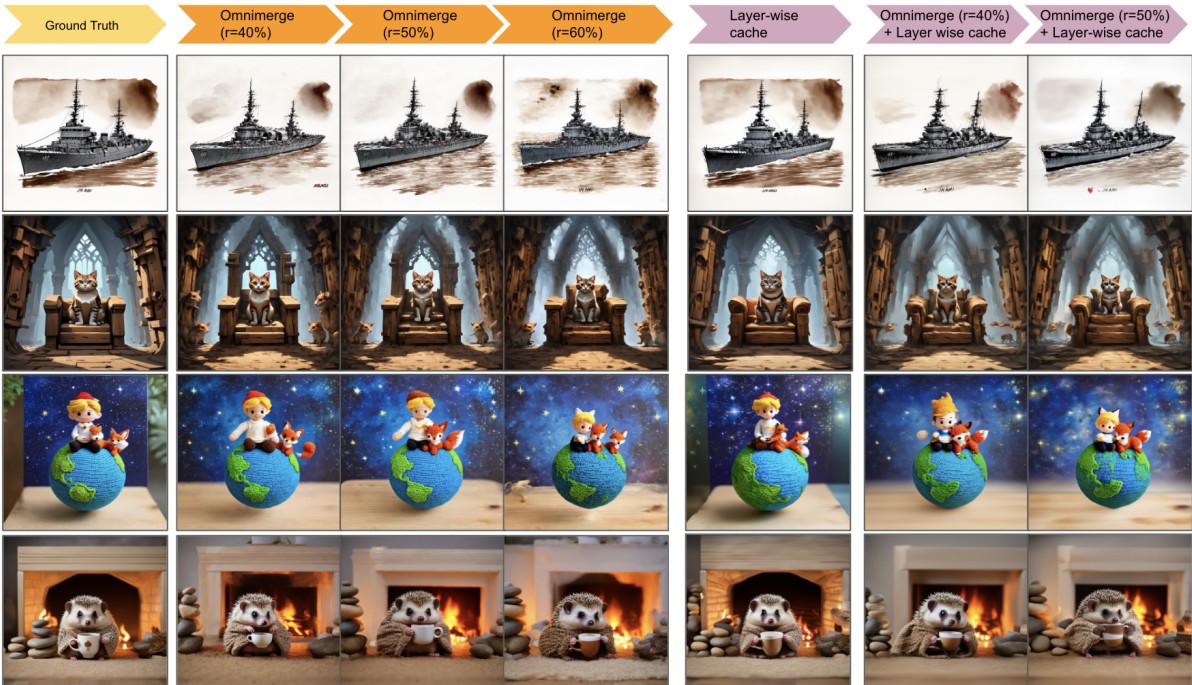

Figure 7: Visualization examples for *OmniCache*, i.e., hierarchical token caching (with use of token cache for unmerging) for merge ratios (40%, 50%, 60%), layer-wise caching, and the integration of both approaches. Prompts used to generate these images are provided in appendix

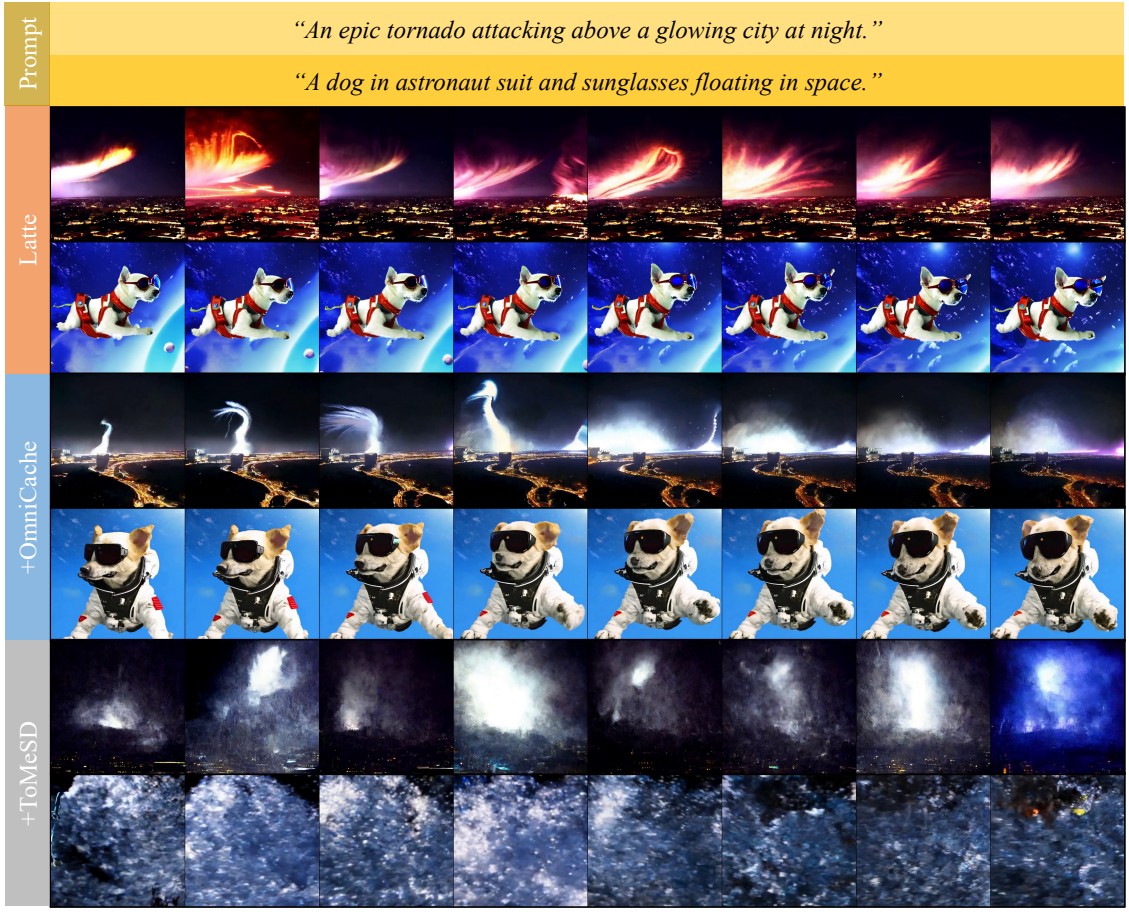

Figure 8: Performance of *OmniCache* on Latte. With $r_f$, $r_b$, $r_t$ set to (30%, 40%, 20%), compared against ToMeSD with $r_t = 25\%$ at the same compression rate. We achieve similar performance to the original Latte while improving inference time by 28%.

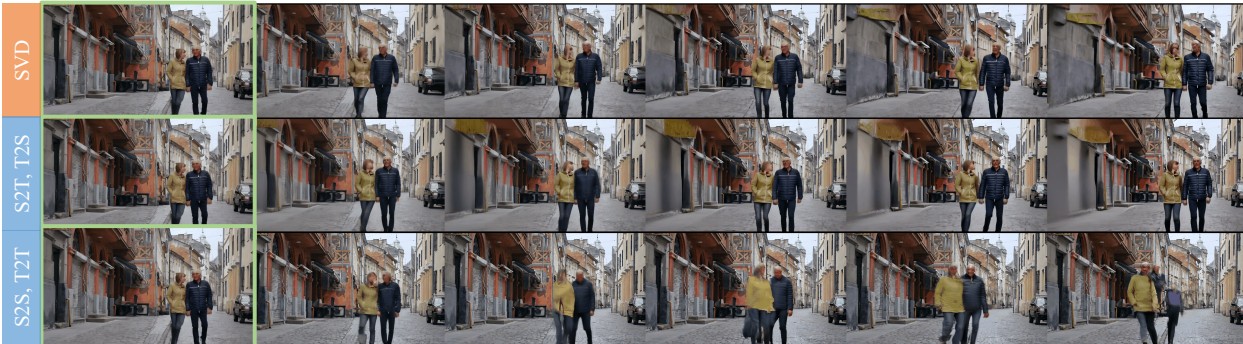

Figure 9: *OmniCache* proposes Spatially compress temporal (S2T) and temporally compress spatial (T2S) vs. Spatially compress temporal (S2S) and temporally compress spatial (T2T). $r_f$, $r_b$, $r_t$ are set to (30%, 30%, 0%). To ensure a fair comparison, Token Merge is not included since it can only be applied in the spatial layer and is irrelevant to this issue. Experiments demonstrate that our proposed S2T and T2S retain more information.

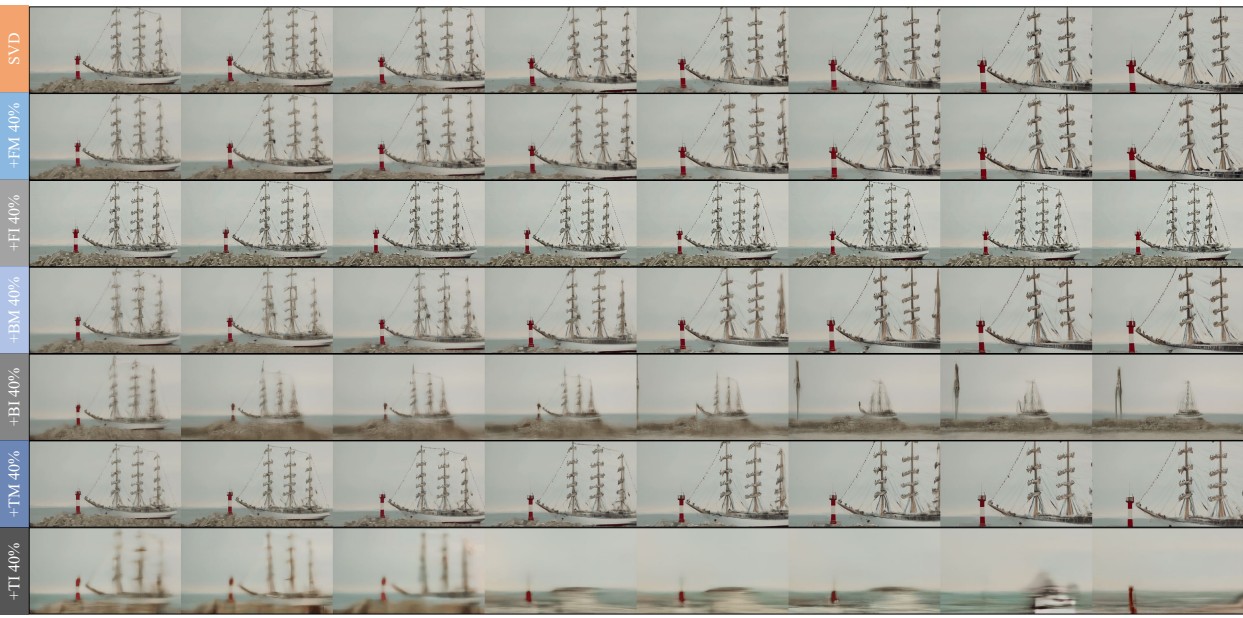

Figure 10: Comparing Frame Cache, Block Cache (BM), and Token Cache (TM) with frame interpolation (FI), block interpolation (BI), and token interpolation (TI). To ensure fairness, all compression ratios are set to 40%. We conduct experiments on the image-to-video model of SVD.

