# OpenReview forum: "OmniCache: Multidimensional Hierarchical Feature Caching for Diffusion Models"
_TMLR — Rejected by TMLR_

### Review · Reviewer_nohT · 2025-12-20

**Summary Of Contributions:**

This paper presents OmniCache, a training-free acceleration framework for diffusion models that exploits four types of redundancy: intra-frame, inter-frame, motion, and step redundancy. The method introduces Frame Cache, Block Cache, Token Cache, and Layered Cache to achieve 25-35% speedup on state-of-the-art models (SD3, SVD-XT, Latte) while maintaining visual quality.

**Additional Comments:**

The work has practical value but needs some revisions to meet publication standards for TMLR. Code availability: Not mentioned. For a systems paper claiming engineering contributions, this is critical.

**Audience:**

Yes

**Audience Explanation:**

Reusable Framework: The four-dimensional redundancy taxonomy could inspire future work in efficient multimodal generation (audio-visual models, 3D generation).
Open Science: If code is released, accelerates research in video synthesis, autonomous driving simulation, medical imaging.

**Broader Impact Concerns:**

Reduced Carbon Footprint: 25-35% inference speedup directly translates to lower energy consumption. For large-scale deployment (e.g., millions of users generating videos daily), this could significantly reduce data center emissions.

Quantification needed: Estimate CO₂ savings. Example: "If applied to 1M daily video generations, OmniCache could save ~X MWh annually, equivalent to Y tons of CO₂.

Lower Hardware Requirements: Training-free acceleration enables video generation on consumer GPUs (e.g., RTX 3090 instead of A100), making creative tools accessible to individual creators and small studios.

Cost Reduction: 25% speedup = 25% lower inference costs for cloud providers, potentially lowering API pricing for end users.

**Claims And Evidence:**

Yes

**Claims Explanation:**

1.Comprehensive redundancy analysis: The four-dimensional redundancy: intra-frame, inter-frame, motion, step wise redundancy provide a clear framework for understanding computational waste in video diffusion models.
2.Training-free approach: No model retraining required makes this practically valuable for deployment.
3.Hierarchical design: The patch-based hierarchical caching (Section 3.3) is clever, reducing overhead by 15-20× while maintaining quality.
4.Broad evaluation: Testing on both UNet-based (SVD-XT) and Transformer-based (Latte, SD3) architectures demonstrates generalizability.
Engineering contribution: Custom Triton kernels show meaningful speedups over PyTorch implementations.

**Requested Changes:**

Add ToCa, Learning-to-Cache, and recent video-specific accelerators.
Limitations section: Acknowledges training potential but doesn't discuss.
When does OmniCache fail? (e.g., high-motion videos)
Compatibility with other acceleration methods (distillation, quantization).

---

> ### Author Response · Authors · 2026-01-13
> **Response on Additional Related Work**
>
> We thank the reviewer for the insightful suggestion and for directing us to relevant work in this research area. Below, we provide a concise discussion of these methods and clarify how they relate to our proposed approach.
>
> ToCa [1] and Learning-to-Cache [2] propose inference-time acceleration methods for transformer-based diffusion models by focusing on local reuse decisions. ToCa selects tokens to cache within transformer layers based on similarity and noise or error sensitivity, while Learning-to-Cache adopts an optimization-based approach to learn a timestep-dependent router that caches entire transformer layers via a differentiable objective. Despite their effectiveness, both methods primarily operate along a single redundancy axis, either token-level or step/layer-level reuse.
> In contrast, our method treats reuse as a **data-scheduling problem with explicit system-level constraints**, coordinating **hierarchical reuse** decisions across multiple redundancy axes—frame, block, token, and denoising steps—under a deterministic budget. This formulation yields a single inference-time control interface that applies to both image and video diffusion backbones without requiring training or learned routers.
>
> In addition, we will reference recent video-specific inference accelerators such as VidToMe [3] and BlockDance [4], which exploit token-level or spatio-temporal redundancy in video diffusion. These approaches typically focus on a single reuse mechanism, whereas our work adopts a broader, multi-axis coordination perspective.
>
> We will revise Section 2.3 (Related Work) of the manuscript to clearly reflect these distinctions and clarify how our framework complements and generalizes existing diffusion acceleration methods.
>
>
> **References**
>
> [1] Zou, C., Liu, X., Liu, T., Huang, S., and Zhang, L. Accelerating Diffusion Transformers with Token-Wise Feature Caching. arXiv preprint arXiv:2410.05317, 2024.
>
> [2] Ma, X., Fang, G., Mi, M. B., and Wang, X. Learning-to-Cache: Accelerating Diffusion Transformers via Layer Caching. In Advances in Neural Information Processing Systems (NeurIPS), 2024.
>
> [3] Li, X., Ma, C., Yang, X., and Yang, M.-H. VidToMe: Video Token Merging for Zero-Shot Video Editing. In Proceedings of the IEEE/CVF Conference on Computer Vision and Pattern Recognition (CVPR), 2024.
>
> [4] Zhang, H., Gao, T., Shao, J., and Wu, Z. BlockDance: Reusing Structurally Similar Spatio-Temporal Features to Accelerate Diffusion Transformers. In Proceedings of the IEEE/CVF Conference on Computer Vision and Pattern Recognition (CVPR), 2025.

---

> > ### Author Response · Authors · 2026-01-13
> > **Response on Limitations and Compatibility with Other Acceleration Methods**
> >
> > We thank the reviewer for raising this important point and agree that a clearer discussion of limitations and interactions with other acceleration techniques would strengthen the paper. OmniCache relies on the presence of redundancy across spatial, temporal, and denoising-step dimensions. In scenarios with extremely rapid or non-repetitive motion (e.g., fast camera shake or dense scenes with independently moving objects), or prompts that require globally varying fine details at every frame, the similarity between cached and current features can decrease. In such cases, aggressive caching configurations (e.g., large $r_f, r_b, r_t$ or short caching intervals $T$ ) may lead to oversmoothing or missed fine details.
> >
> > Our ablation results (Table 3) indicate that conservative settings, such as frame caching $\le 40 $%, token caching $\le 40$%, and block caching $\le 30$% provide a robust operating regime across diverse inputs, while more aggressive configurations may introduce artifacts. We will clarify these regimes and recommend conservative default settings in the revised manuscript.
> >
> > We also emphasize that OmniCache is orthogonal to many existing acceleration techniques. Step-reduction or distillation-based methods reduce the number of denoising steps, whereas OmniCache reduces per-step computation via feature reuse, suggesting that the two can be combined for multiplicative speedups. Similarly, quantization and low-precision kernels reduce the cost of individual operations, while OmniCache reduces the number of operations by reusing intermediate representations. One potential interaction is that heavy quantization may slightly perturb similarity estimates; in practice, this can be mitigated by using more conservative caching ratios or computing similarity metrics at higher precision. We will add a discussion of these interactions and leave a comprehensive empirical study of combined methods to future work.

---

> ### Author Response · Authors · 2026-01-13
> **Response on Broader Societal Impact**
>
> We thank the reviewer for highlighting the broader societal implications of inference-time efficiency. Since OmniCache is a training-free, inference-only optimization that reduces wall-clock latency without modifying the model architecture, energy consumption per generation scales approximately linearly with the achieved speedup for a fixed hardware configuration.
>
> As a concrete example, in our SVD-XT setup the latency for generating a 25-frame video at 576 × 1024 resolution decreases from 110 s to 82.4 sec (Table 3), corresponding to a 25% reduction. Assuming a single 300 W GPU, the baseline energy per generation is approximately $E_{base} \approx 300W \times 110s/3600 \approx 9.2$Wh, while OmniCache consumes $E_{omni} \approx 300 W \times 82.4s/3600 \approx 6.9 $Wh. This yields an energy saving of about 2.3 Wh per video ($ \approx 25 $%). At a scale of one million video generations per day, this corresponds to approximately 2.3 MWh saved per day, or about 840 MWh annually. Using a conservative grid emission factor of 0.4 tCO₂/MWh, this translates to roughly 336 tons of CO₂ avoided per year. While exact values depend on hardware efficiency and data-center energy sources, this example illustrates that a 25–35% inference speedup can lead to material reductions in energy consumption and associated emissions at scale.
>
> Beyond environmental impact, OmniCache can also lower hardware and deployment barriers. Because it operates purely at inference time and requires no retraining or finetuning, it can be applied directly to pretrained diffusion models on commodity GPUs (e.g., RTX 3090/4090), enabling faster video generation with reduced memory and compute overhead. This makes advanced video diffusion models more accessible to individual creators and small studios with limited hardware budgets.
>
> For cloud deployments, inference speedups translate almost linearly into reduced GPU-hour consumption. In our experiments, OmniCache reduces latency by approximately 25% on SVD-XT, 28% on Latte, and 35% on SD3 (Sections 4.3 and 6). At fixed throughput, this corresponds to requiring 25–35% fewer GPU-hours. For example, a service spending 1,000 GPU-hours per day on video diffusion inference could reduce usage to approximately 750 GPU-hours per day with a 25% speedup. Such reductions can lower infrastructure costs for providers and potentially be reflected in reduced API pricing for end users.

---

> ### Author Response · Authors · 2026-01-13
> **Availability of Code**
>
> We agree with the reviewer that code availability is important for reproducibility and for evaluating the engineering contributions of this work. We will release the full implementation of OmniCache, including inference scripts and configuration files, upon acceptance of the paper, and will include a clear link in the revised manuscript.

---

### Review · Reviewer_2fxT · 2025-12-26

**Summary Of Contributions:**

This paper introduces OmniCache for accelerating diffusion models by reducing four types of redundancies: intra-frame, inter-frame, motion, and step. The proposed method integrates multiple caching modules, including Token Cache, Frame Cache, Block Cache, Layered Cache, enabling training-free speedup across both image and video diffusion models.

Strengths:
Systematic and unified design covering multiple redundancy types. Strong empirical results on multiple SOTA models (SD3, SVD-XT, Latte). Practical and deployment-friendly. Detailed ablation and kernel optimization for reproducibility.

Weaknesses:
Theoretical insights are limited, as method is mostly engineering-driven. Caching strategy search is heuristic and could be more principled.

**Audience:**

Yes

**Audience Explanation:**

The paper addresses a critical bottleneck in diffusion models: inference efficiency. Its training-free, architecture-agnostic design makes it highly relevant to researchers and practitioners working on scalable generative models, especially in video generation and deployment scenarios.

**Broader Impact Concerns:**

No major ethical concerns.

**Claims And Evidence:**

No

**Claims Explanation:**

While the empirical results are extensive and convincing, some claims (e.g., optimality of the caching strategy) rely on heuristic algorithms without strong theoretical justification. The method is well-engineered, but lacks formal analysis for its design choices. Strengthening theoretical support would improve credibility.

**Requested Changes:**

1. Clarify theoretical motivation: While the design is intuitive, the paper would benefit from more formal analysis or justification for why the proposed caching strategy is effective beyond empirical results.

2. Improve caching strategy section: The current approach is heuristic. Discuss the potential of more principled scheduling methods (e.g., reinforcement learning or optimization-based scheduling).

---

> ### Author Response · Authors · 2026-01-13
> **Response on Theoretical Motivation and Scheduling Strategy**
>
> We thank the reviewer for the thoughtful feedback and agree that clarifying the theoretical motivation behind our design choices would strengthen the paper. While OmniCache is a training-free inference-time system, its design is guided by a principled view of feature reuse as an **adaptive resource-allocation problem** across multiple redundancy axes: intra-frame (token-level), inter-frame (frame-level), motion (block-level), and denoising-step redundancy. Rather than caching uniformly, OmniCache explicitly estimates where redundancy is highest and where the model is most tolerant to approximation.
>
> Concretely, for the Layered Cache component, given a target speedup $s_r$ and cache interval $T$, we associate each layer-timestep pair $(l,t)$ with a weight $w(l,t)$ = $\Delta$quality$(l,t)$/latency$(l,t)$, which measures quality degradation per unit of saved latency. Cache positions are then selected greedily based on this *benefit-per-cost* ratio, followed by a dependency-aware refinement step based on the model's compute graph to respect execution constraints (Section 3.5, Algorithms 1-3). This procedure can be interpreted as a **knapsack-style, budget-constrained scheduling** that generates cache schedules that respects both model's architecture and diffusion timesteps without any retraining.
>
> For spatial and temporal caching, our design is motivated by the role separation in modern video diffusion models: temporal layers primarily model motion and are relatively insensitive to moderate spatial subsampling, while spatial layers primarily model appearance and are more tolerant to moderate temporal subsampling. This observation is supported by our S2T/T2S vs. S2S/T2T experiments (Figure 9), which shows that compressing layers along their secondary dimension preserves quality, whereas compression along their primary dimension leads to noticeable artifacts. Toghether, these observations provide a conceptual justification for why combining Frame/Block/Token-level caching is effective beyond empirical performance alone.
>
> We also agree that more formal scheduling approaches are an interesting direction for future work. While OmniCache already adopts a budget-based optimization perspective: it estimates per-layer latency and quality degradation and constructs cache schedules via greedy selection followed by graph-level refinement. Learning-based or optimization-based schedulers (e.g., reinforcement learning or black-box optimization) could potientally learn input/prompt-dependent cache policies that can trade-off speed and quality per input or maximize end-to-end rewards (e.g., FVD, FID) based on specific latency budget. Importantly, our hierarchical design exposes discrete, interpretable control variables ($r_f$, $r_b$, $r_t$, $T$, and layer selection), making OmniCache naturally compatible with such extensions. We view these approaches as complementary rather than prerequisites for the current system.
>
> In the revised manuscript, we will clarify the theoretical motivation of the caching strategy, explicitly articulate the intutions underlying multi-axis reuse, and position more formal scheduling approaches as complementary future directions.

---

### Review · Reviewer_CxLD · 2026-01-01

**Summary Of Contributions:**

This paper proposed OmniCache, which is a training-free framework to accelerate the inference of large-scale image and video diffusion models by resolving four kinds of redundancy (intra-frame, inter-frame, motion, and step redundancy). The main behind OmniCache is based on the hierarchical patch-based caching, which partitioned features into patches and applied caching in a path-wise manner. Empirical evaluations on a wide range of state-of-the-art (SOTA) models like Stable Diffusion 3 (SD3) and Stable Video Diffusion (SVD) are included to justify the effectiveness of the proposed methodology.

**Additional Comments:**

NA

**Audience:**

Yes

**Audience Explanation:**

To the best of the reviewer's knowledge, efficient inference of large-scale image/video diffusion models (especially Stable Diffusion and its variants) is a hot topic in both academia and industry. Moreover, the hierarchical caching design proposed in this paper may also inspire further work on multi-axis redundancy exploitation in diffusion models. Hence, the reviewer does think that this paper will be of interest to the TMLR community.

**Claims And Evidence:**

Yes

**Claims Explanation:**

It seems to the reviewer that the central claim (OmniCache can speed up inference by approximately 25 to 35 percent on SVD-XT, Latte, and SD3-medium with minor quality degradation) is reasonably supported by the numerical experiments included in the manuscript. For instance, Tables 2,3,4 and Figures 6,7,8 together exhibit consistently better FVD/FID/CLIP scores and speedups compared to the no-caching baselines.

**Requested Changes:**

Overall, the reviewer thinks that this paper is well-engineered and practically useful, but the authors probably need to take some efforts to further explain the conceptual novelty of the paper. Specifically, it seems that each component of the current methodology is quite close to existing ideas (DeepCache for stepwise caching, ToMe/ToMeSD for similarity-based token merging, etc). Would it be possible for the authors to further emphasize the main novelty of the work (how current ideas for resolving redundancies are effectively combined) and add it to the "core contributions" section of the work? This will be essential for me to recommend acceptance.

---

> ### Author Response · Authors · 2026-01-12
>
> ## Response to Reviewer CxLD
>
> We sincerely thank the reviewer for the positive assessment and the constructive suggestion regarding the clarification of **conceptual novelty**. We fully agree that, beyond strong empirical results, it is important to clearly articulate *what is fundamentally new* in OmniCache compared to prior caching and token-merging approaches. Below we provide a concise clarification and outline concrete revisions we will make.
>
> ---
>
> ### Clarifying the Novelty of OmniCache
>
> While some *individual operations* in OmniCache may resemble existing ideas (e.g., step-wise caching in DeepCache or token merging in ToMe/ToMeSD), the novelty of OmniCache lies in a **holistic, redundancy-driven system design** that is fundamentally different from prior single-axis methods. We will emphasize the following key contributions more explicitly in the revised manuscript.
>
> #### 1. Redundancy-first analysis tailored to *video diffusion*
> OmniCache is built on a unified analysis of **four distinct but complementary redundancies** in diffusion-based image and video generation:
> - **Intra-frame redundancy** (spatial repetition within a frame),
> - **Inter-frame redundancy** (high similarity across adjacent frames),
> - **Motion redundancy** (sparse, block-wise temporal motion),
> - **Step redundancy** (slow feature evolution across denoising steps).
>
> Unlike prior work that targets only one redundancy type, this analysis directly guides *where* and *how* caching is applied in the model.
>
> #### 2. Structure-aware caching aligned with spatial–temporal layers
> A central design insight of OmniCache is that **the effectiveness of caching depends on matching redundancy types to the architectural roles of layers**:
> - We **cache spatial information inside temporal layers** to exploit spatial structure while preserving motion sparsity.
> - We **cache temporal information inside spatial layers** to exploit frame similarity without collapsing motion dynamics.
>
> Our experiments (e.g., S2T/T2S vs. S2S/T2T) show that mismatched compression leads to significant quality degradation, highlighting a design principle that is absent from prior token merging or step caching methods.
>
> #### 3. Unified multi-granularity caching framework
> OmniCache is, to our knowledge, the **first framework to jointly integrate caching at four orthogonal granularities**:
> - **Token Cache** (token-level, intra-frame),
> - **Block Cache** (block-level, motion),
> - **Frame Cache** (frame-level, inter-frame),
> - **Layered Cache** (layer × timestep, step redundancy).
>
> These components are not independent heuristics but are **hierarchically coordinated**, enabling flexible allocation of compression across space, time, tokens, and layers. This unified design consistently yields a better efficiency–quality tradeoff than prior methods, as demonstrated in Tables 2–4 and Figures 6–8, even when compared with strong baselines such as DeepCache, layer caching, and ToMeSD under similar budgets.
>
> #### 4. Hierarchical caching for real GPU speedups
> Beyond algorithmic design, OmniCache addresses a key practical challenge: **naïve caching often fails to translate into wall-clock speedups under modern FlashAttention-style kernels**. We introduce:
> - Patch-wise hierarchical caching to drastically reduce similarity-computation overhead,
> - Layer- and timestep-aware scheduling to align caching decisions with GPU execution efficiency,
> - Custom Triton kernels to fuse merge/unmerge operations efficiently.
>
> This combination enables *actual inference acceleration* in real deployments, not just theoretical FLOP reductions, which we believe is an important and underexplored contribution.
>
> ---
>
> ### Planned Revisions
>
> To address the reviewer’s suggestion, we will make the following updates in the revised manuscript:
>
> - **Introduction & Core Contributions**
>   Rewrite the novelty statement to explicitly emphasize the redundancy-driven, multi-granularity, and structure-aware nature of OmniCache.
>
> - **Related Work**
>   Expand the discussion of DeepCache, Δ-DiT, ToMe, and ToMeSD, clearly contrasting their single-axis designs with our unified hierarchical framework.
>
> - **Method Section**
>   Add a concise paragraph summarizing how each design choice (Frame, Block, Token, and Layered Cache) corresponds to a specific redundancy and architectural role.
>
> We believe these revisions will make the conceptual contribution of OmniCache clearer and more compelling, and we thank the reviewer again for highlighting this important point.

---

> ### Author Response · Authors · 2026-01-12
>
> ## Additional Response on Related Work Clarification
>
> We thank the reviewer for highlighting closely related acceleration methods and for encouraging a more precise positioning of our work within this line of research.
>
> Conceptually, **ToCa** [1] and **Learning-to-Cache** [2] both focus on *local reuse decisions* within diffusion transformers.
> ToCa selects tokens to cache and reuse inside transformer layers based on token similarity and noise/error sensitivity, while Learning-to-Cache learns a timestep-dependent routing policy to cache entire transformer layers via differentiable optimization. Despite their effectiveness, both methods primarily operate along a **single redundancy axis**—either token-level or step/layer-level reuse.
>
> In contrast, **OmniCache formulates caching as a global, hierarchical resource-allocation problem**. Rather than optimizing reuse decisions independently, OmniCache *jointly coordinates caching across multiple orthogonal axes*—**frame-level, block-level, token-level, and denoising-step-level**—under an explicit, deterministic, budget-constrained optimization. This design enables a **single inference-time control interface** that naturally generalizes across both image and video diffusion backbones, without requiring training or learned routers.
>
> We will also explicitly reference recent **video-specific inference acceleration methods**, including **VidToMe** [3] and **BlockDance** [4]. These approaches exploit token-level or spatio-temporal block-level redundancy in video diffusion models. While effective, they are each centered on a *single reuse mechanism* (token merging or block similarity). OmniCache, by contrast, is positioned as a **unifying framework** that integrates these reuse principles across *multiple granularities and architectural roles*, enabling more flexible tradeoffs between efficiency and quality.
>
> In the revised manuscript, we will update **Section 2.3 (Related Work)** to clearly articulate these distinctions and to emphasize how OmniCache complements, subsumes, and generalizes existing diffusion acceleration methods along different redundancy dimensions.
>
> **References**
>
> [1] Zou, C., Liu, X., Liu, T., Huang, S., & Zhang, L. (2024). *Accelerating diffusion transformers with token-wise feature caching*. arXiv:2410.05317.
> [2] Ma, X., Fang, G., Bi Mi, M., & Wang, X. (2024). *Learning-to-cache: Accelerating diffusion transformer via layer caching*. NeurIPS.
> [3] Li, X., Ma, C., Yang, X., & Yang, M. H. (2024). *VidToMe: Video token merging for zero-shot video editing*. CVPR.
> [4] Zhang, H., Gao, T., Shao, J., & Wu, Z. (2025). *BlockDance: Reuse structurally similar spatio-temporal features to accelerate diffusion transformers*. CVPR.

---

### Author Response · Authors · 2026-01-20
**Revised Manuscript Uploaded**

We would like to thank the reviewers for their thoughtful and constructive feedback. We have uploaded a revised version of the manuscript that addresses the reviewer's concerns, including clear articulation of the conceptual novelty of *OmniCache*, expanded and a better-positioned related work, and an improved discussion of limitations and applicability. The reviewers’ suggestions were extremely helpful in improving both the technical clarity and overall presentation of the paper.

---

### Decision · Action_Editor_8a8w · 2026-03-01

**Recommendation:** Reject

**Additional Comments:**

The authors should clearly clarify whether the main contribution lies in identifying new forms of redundancy in diffusion models or in proposing an engineering optimization for specific architectures. They should more explicitly distinguish their method from prior work such as DeepCache and ToMe/ToMeSD, especially since the individual components are highly similar. If the contribution is the effective combination of these components, the paper should articulate what new methodological insight or principle emerges from this integration. The authors should also provide stronger evidence that the approach generalizes across different models and scenarios, rather than reflecting architecture-specific tuning.

**Audience:**

Yes

**Audience Explanation:**

Some individuals in TMLR’s audience, particularly those working on efficient diffusion models and inference acceleration, would likely be interested in the empirical findings and practical engineering insights.

**Claims And Evidence:**

No

**Claims Explanation:**

The claims are only partially supported by clear and convincing evidence. The empirical results show practical effectiveness on the evaluated models. However, the submission does not provide sufficient theoretical justification or principled analysis to substantiate its claims of novelty and a general training-free framework. The evidence supports engineering improvements in specific settings, but it does not convincingly establish broader conceptual contributions or generalizability.

**Resubmission Of Major Revision:**

The authors may consider submitting a major revision at a later time.